# Study of Concrete Strength and Pore Structure Model Based on Grey Relation Entropy

**DOI:** 10.3390/ma14020432

**Published:** 2021-01-16

**Authors:** Min Zhang, Xianhua Yao, Junfeng Guan, Lielie Li, Juan Wang, Longbang Qing

**Affiliations:** 1School of Civil Engineering and Communication, North China University of Water Resources and Electric Power, 36 Beihuan Road, Zhengzhou 450045, China; zhangmin@ncwu.edu.cn (M.Z.); yaoxianhua@ncwu.edu.cn (X.Y.); 100--lilili@163.com (L.L.); 2School of Water Conservancy and Engineering, Zhengzhou University, 100 Science Avenue, High-Tech Development Zone, Zhengzhou 450001, China; 3College of Civil Engineering, Hebei University of Technology, 5340 Xiping Road, Beichen District, Tianjin 300401, China; qing@hebut.edu.cn

**Keywords:** concrete strength relationship, pore structure, mercury intrusion porosimetry, grey relation entropy (GRE), sensitivity pore size

## Abstract

The Grey Relation Entropy (GRE) theory is used to analyze the sensitive pore size that affects the compressive strength of concrete. The relationship between the strength and pore structure is revised based on the sensitivity coefficient. The revised model is used to calculate the compressive strength of concrete. In order to verify the validity of the proposed model, the calculated results are compared with experimental ones, showing satisfactory agreement with a larger correlation than existing methods.

## 1. Introduction

Concrete is a composite material with a porous structure, which affects its strength, impermeability and durability, but mostly its strength [1,2]. The macroscopic strength of concrete is closely related to its microscopic pore structure. Back in 1960, Powers [3] presented the theory of gel-to-air ratio, which linked the strength of concrete with the volume of capillary pores and concluded that the number of pores in concrete and the pore size range are related to the initial water–cement ratio and the hydration degree of the cement. Based on that strength theory, Balshin [4], Schiller [5], Hansen [6], Ryshkewitch [7], Guo and Wu [8], as well as Jons and Osbaeck [9], proposed different formulas that related the concrete’s strength to porosity. However, these formulas consider only the influence of porosity on the concrete’s strength, neglecting all other influencing factors. Pann and Yen [10] studied the influence of water–cement ratio, porosity and hydration degrees on the concrete’s strength, and proposed a formula that connected all these characteristics. Kollas [11] dried concrete specimens in the saturated state, measured the porosity by mercury intrusion porosimetry, introduced the concept of average pore size, and proposed a new formula between concrete strength and average porosity. Jambor [12] proposed that the pore size distribution is an important factor affecting the strength of concrete. The test results showed that, considering a constant porosity, the smaller the pore size, the larger the strength of concrete. Wu and Lian [13] considered the two factors of comprehensive pore gradation and porosity to establish a strength model; they introduced the concept of subporosity and improved the concrete performance by optimizing the pore structure. Huang [14] studied the concrete strength at the macroscopic, mesoscopic and molecular levels and specified the three factors that have a significant impact on the concrete’s strength: stress concentration, cement stone and pore structure. Older and Rößler [15] proposed a linear relationship between strength and porosity of different pore sizes, while Atzeni et al. [16] modified this relationship. However, the modified linear formula contains many fitting parameters. Wittmann [17] introduced the elastic modulus, the porosity of concrete and the fracture energy parameter of fracture mechanics into the strength formula. Atzeni [16] also included porosity and average pore size into the relationship between concrete strength and pore structure. Facing the problem of parameter determination, they suggested that, when the porosity is equal to zero, the concrete strength value is determined by a linear formula. Kumar and Bhattacharjee [18] proved that the uniaxial compressive strength of a material depends on its tensile breaking strength. Based on the strength formula proposed by Atzeni et al. [16] and the critical tensile strength formula proposed by Tang [19], Kumar and Bhattacharjee [18] established the relationship between the cement content, porosity, average pore size and concrete strength, combined with the classic Griffith fracture theory. However, this formula does not consider pore size distribution factors that influence the concrete strength.

Because of the complexity of concrete composition, the factors affecting its strength are various. In order to establish the relationship between concrete strength and pore structure, the grey correlation analysis method has been applied the main factors that affect the system [20,21,22]. However, this method has the disadvantages of partial correlation tendency and loss of individual information, which cause significant deviation of the grey correlation degree and affects the analysis results [23,24]. Mokhtar et al. [25], Jin [26], Chen [27], Xie et al. [28], used the Grey Relational Entropy (GRE) analysis method to study the influencing factors and laws of concrete strength, which was proved to be a simple and effective method.

Therefore, based on the GRE theory, this article analyzes the most sensitive factors that affect the strength of concrete. Considering the most influencing factors, such as cement content, porosity, average pore size and sensitive pore size distribution, the quantitative relationship between the pore structure of concrete and its compressive strength is proposed. Hence, this paper provides a new method for establishing the connection law between the microstructure and macroscopic properties of concrete.

## 2. Analysis of Sensitive Factors of Concrete Compressive Strength

### 2.1. GRE Theory

In order to solve the problem of partial correlation tendency and loss of individual information, which cause significant deviation of the grey correlation degree and affect the analysis results, based on grey correlation analysis, GRE theory is introduced to quantitatively describe the relation degree between various factors. The calculation theory is as follows.

The reference sequence is *X*_0_(*k*) and the comparative data column is *X_i_*(*k*)(*k* = 1,2, … *m*; *i* = 1,2, … *n*). Firstly, equalization with *X*_0_(*k*) and *X_i_*(*k*) is taken into account based on the formula $Y(k)=nX(k)/\sum_{i=1}^{n}X(k)$. Then, the sequence *Y*_0_(*k*) and *Y_i_*(*k*) is obtained. The correlation coefficient is calculated according to Equation (1) [29,30]:(1)$$R_{i}(k)=\frac{\underset{i}{\min}\left|Y_{0}(k)-Y_{i}(k)\right|+\rho \underset{i}{\max}\left|Y_{0}(k)-Y_{i}(k)\right|}{\left|Y_{0}(k)-Y_{i}(k)\right|+\rho \cdot \underset{i}{\max}\left|Y_{0}(k)-Y_{i}(k)\right|}$$
where min and max represent the minimum and maximum values of the difference which is $\left|Y_{0}(k)-Y_{i}(k)\right|$; *ρ* is the resolution coefficient, generally 0.5.

The value of GRE for correlation coefficient series *R_i_*(*k*) is calculated by Equation (2) [29,30].
(2)$$H(R_{i})=-\sum_{h=1}^{m}P_{h}\ln P_{h}$$
where *H*(*R_i_*) is the value of GRE; *P_h_* is the density value of the mapping distribution of correlation coefficient series *R_i_(k*), $P_{h}=R_{i}(h)/\sum_{k=1}^{m}R_{i}(k)$, *h* = 1,2, … *m*.

The relation degree of GRE *D*(*x_i_*) is calculated by Equation (3) [29,30].
*D*(*x_i_*)= *H*(*R_i_*)/H_max_(3)
where H_max_ is the maximum grey relation entropy and H_max_ = ln*m*, *m* is the number of data elements.

### 2.2. Sensitivity Analysis of Pore Structure Parameters Based on GRE Method

Concrete porous structure parameters include: total pore volume *V*_total_, average pore radius *r_m_*, porosity *P*, distribution rate of pore radius *B*, and the most probable pore radius *r*_kj_. The GRE method was used to select the most sensitive factors that affect the strength of concrete in order to find the relationship between them.

The mercury intrusion porosimetry (MIP) method was used to test the pore characteristics of concrete materials; this method can accurately detect the porosity, pore radius distribution and other pore properties [31]. Hence, the GRE analysis was performed on the concrete mercury intrusion test data, denoted as data I, which are given in [18], and shown in Table 1.

The concrete strength was used as the target magnitude, and 7 indicators were taken into account as the comparison sequence, which are the cement content *C*, the average pore radius *r_m_*, the porosity *P*, and the distribution rate of four different pore radii. The values of corresponding relation degree of GRE are shown in Table 2.

The values of relation degree of GRE, calculated by Equation (3) and listed in Table 2, are close, following the descending order: *D*(*x*_5_) > *D*(*x*_3_) > *D*(*x*_6_) > *D*(*x*_4_) > *D*(*x*_7_) > *D*(*x*_1_) > *D*(*x*_2_). The sensitive pore radius is the pore radius corresponding to the factor with the largest sensitivity coefficient. The maximum value of the relation degree is *D*(*x*_5_)—thus, the distribution rate of 10.6–53 nm pore radius is the most sensitive. Based on the above values, it was noted that the cement content can also significantly influence the concrete strength, so it should also be included in the strength formula.

The pore structure parameter data of concrete, mixed with 9 kinds of air-entraining agents, named as data II, were retrieved from [21] and are shown in Table 3.

In the same way, the GRE analysis was performed on the test data II. The concrete strength was used as the target magnitude and 8 indicators were considered as the comparison sequence, which are the average pore radius *r_m_*, porosity *P* and distribution rate of six different pore radius.

The values of relation degree, shown in Table 4, are close too—in descending order: *D*(*x*_6_) > *D*(*x*_2_) *> D*(*x*_3_) *> D*(*x*_7_) *> D*(*x*_5_) *> D*(*x*_8_) *> D*(*x*_1_) *> D*(*x*_4_). In this case, the maximum value of relation degree is *D*(*x_6_*), thus the distribution rate of 600–800 µm pore radius is the most sensitive.

Based on the GRE method analysis results of the test data I and data II, it was found that: (1) the cement content is a significant influencing factor for concrete strength; (2) among the concrete pore structure parameters, the pore radius distribution rate is considerably sensitive for the calculation of concrete macroscopic strength; (3) for different types of concrete, the sensitive pore radius that affects the strength of concrete is different, and it needs to be determined by GRE method analysis according to the pore radius distribution rate in the micropore structure parameters.

## 3. Revise Compressive Strength-Pore Structure Model 

### 3.1. Sensitive Pore Size Distribution Rate B_s_ and Sensitivity Coefficient ƞ

Based on the analysis of sensitive factors, it can be noted that the distribution rate of pore radius has a greater influence on the concrete’s compressive strength than other factors in the microporous structure. The sensitive pore is the pore corresponding to the factor with the largest sensitivity coefficient. Because the radii of sensitive pores are within a range, the average value of the pore radii is the average pore radius of sensitive pores. In the following it is assumed that the cross-sectional area of concrete sample is *A*, the number of pores is *n*_0_, the pore area is *A_0_*, the porosity is *P*, the average pore radius is *r_m_*, the number of sensitive pores is *n*_s_, the average pore radius of sensitive pores is *r_ms_*, and the sensitive pore area is *A_s_*. The sensitive pore radius distribution rate is *B_s_* which is defined as the ratio of the sensitive pore area to the pore area, *B_s_* = *A_s_*/*A_0_*. Taking into account that *A_0_* = *n_0_*π*r_m_^2^* and *A_S_* = *n_s_*π*r_ms_^2^*, the calculation of the sensitive pore radius distribution rate is shown in Equation (4).
(4)$$B_{s}={\left(\frac{r_{m_{S}}}{r_{m}}\right)}^{2}\frac{n_{s}}{n_{0}}$$

The sensitivity coefficient *ƞ =*
*r_ms_*/*r_m_* was defined as the ratio of the average pore radius of sensitive pore *r_ms_* over the average pore radius *r_m_*. This coefficient was used to measure the geometric characteristics of sensitive pores, directly reflecting relative size that the sensitive void relative to the total void. The test constant *α* was also defined as *α* = (*n*_0_/*n_s_*)^1/2^, with *α* > 1, which can be obtained by fitting test data. Replacing *ƞ* and *α* into Equation (4), Equation (5) is formed, in which it is noted that the sensitivity coefficient *ƞ* is proportional to the arithmetic square root of the sensitive pore radius distribution rate *B_s_*.
(5)$$\eta =\alpha \sqrt{B_{s}}$$

It should be noted that the sensitive pore radius is the pore radius corresponding to the factor with the largest sensitivity coefficient which is obtained through GRE analysis. Thus, the sensitive pore radius has the greatest influence on the strength of the concrete. Equations (4) and (5) are formulas based on certain assumptions that each pore is an independent circle in a certain section of concrete. 

### 3.2. Revised Strength-Pore Structure Model

Kumar and Bhattacharjee [18] quantitatively studied the relationship between the pore structure and the concrete strength through two pore structure parameters, porosity *P* and average pore radius *r_m_*, also introducing the cement content parameter, denoted as *C*, and a test constant *K*_2_, which can be obtained by fitting test data, as shown in Equation (6) [18].
(6)$$\sigma =K_{2}\frac{C\left(1-P\right)}{\sqrt{r_{m}}}$$
where *K*_2_ is a test constant which is mainly related to the strength of the concrete matrix and is related to factors such as the type of aggregate, the type of cement and the cement content.

In this paper, the influence of the sensitive pore radius is taken into account, introducing the sensitivity coefficient *ƞ*, which is used as evaluation index, into the concrete relationship defined by Equation (6). Equation (6) is revised and a new strength-pore structure model is proposed, as given by Equation (7).
(7)$$\sigma =K_{2}\eta \frac{C\left(1-P\right)}{\sqrt{r_{m}}}$$

Substituting Equation (5) into Equation (7), Equation (8) is obtained.
(8)$$\sigma =\alpha K_{2}\sqrt{B_{s}}\frac{C\left(1-P\right)}{\sqrt{r_{m}}}$$

Based on Equation (8), it was noted that the compressive strength of concrete can be determined by the cement content, porosity, average pore radius and sensitive pore radius distribution rate.

Of note, Equation (8) proposed in this paper firstly performs GRE analysis to find the sensitive pore size, which leads to certain limitations in the use of Equation (8). The use of Equation (8) requires that the pore structure data must meet conditions that the pore radius range is consistent and the concrete strength grade and age should also be consistent.

## 4. Verification of the Revised Model

### 4.1. Common Strength-Pore Structure Model

Balshin [4], Schiller [5], Ryshkewitch [6], and Hansen and Jons [7] proposed several relationships between strength and porosity.

The linear relationship is given by Equation (9) [4].
(9)$$\sigma =\sigma_{0}\left(1-NP\right)$$

The power exponent relationship is given by Equation (10) [5].
(10)$$\sigma =\sigma_{0}{\left(1-P\right)}^{n}$$

The exponential relationship is given by Equation (11) [6].
(11)$$\sigma =\sigma_{0}\exp(-BP)$$

The logarithmic relationship is given by Equation (12) [7].
(12)$$\sigma =k\ln(\frac{\sigma_{0}}{P})$$

Atzeni et al. proposed the strength and pore structure relationship, as shown in Equation (13) [16].
(13)$$\sigma =K\frac{\sigma_{0}\left(1-P\right)}{\sqrt{r_{m}}}$$

In above formulas: *σ*_0_ is the compressive strength when porosity is zero; *K, N*, *B*, *k* and *n* are test constants.

### 4.2. Verification of the Revised Model with Data I

In order to compare the common strength-pore structure models with the revised strength model described by Equation (8), the 36 groups of test data I are considered and analyzed. The results based on the fitting regression analysis are shown in Table 5.

It should be noted that Equations (9)–(12), which consider only the impact of porosity on the concrete’s strength, the correlation coefficient reaches 0.535 on average; Equation (13), considering the factors of average pore radius and porosity, has a correlation coefficient of 0.578; Equation (6), taking into account the average pore radius, porosity, and cement content, has a correlation coefficient of 0.780; Equation (8) has the largest fitting accuracy, reaching 0.850, and the concrete’s strength is related not only to porosity, but also to average pore radius, cement content and sensitive pore radius. Considering the influence of the sensitive pore radius on the strength of concrete, it is more reasonable to predict the strength of concrete with Equation (8) proposed in this paper.

The fitting results are also shown in Figure 1 and Figure 2, respectively.

### 4.3. Verification of the Revised Model with Data II

In order to verify the applicability of the proposed model, test data II was also used. The fitting results of the strength model proposed by Kumar and Bhattacharjee [18] (Equation (6)) and the revised one (Equation (8)) are shown in Figure 3 and Figure 4, respectively.

In Figure 3 and Figure 4, it is shown that, considering the distribution rate of sensitive pore radius in the revised model, the data are more concentrated on both sides of the fitting line, and the regression line has a better correlation coefficient (R = 0.78). Moreover, compared with the correlation coefficient of the original model, it is increased by 18.2%.

Through the analysis of test data II, the test constants and correlation coefficients are obtained by fitting; the strength values of the model and the test strength are shown in Table 6.

## 5. Conclusions

Based on the GRE theory, the relationship between concrete compressive strength and microporous structure parameters was studied. The conclusions are as follows.

(1)Based on the GRE theory, it was concluded that the most sensitive factors that affect the compressive strength of concrete is the distribution rate of the pore radius with the maximum value of relation degree.(2)A new strength and pore structure model is proposed, considering the influence of sensitive pore radius and introducing the concept of sensitivity coefficient.(3)The revised model proposed in this paper and the existing strength model, used to calculate the compressive strength of concrete, are compared with test results, which shows that the revised model and the test results are in better agreement.(4)In the revised model, the concrete’s strength is related not only to porosity, average pore radius and cement content, but also to sensitive pore radius. The consideration of the influence of the sensitive pore radius on the strength of concrete leads to a better estimation of the concrete’s strength.(5)This paper only considers the influence of the sensitive pore radius on the strength of concrete, but does not consider the influence of capillary pores, gel pores, location of pores, and interconnectivity of pores on concrete strength. In the future, the author intends to conduct a porosity test on studied concrete, taking into account the influence of the pore structure on the permeability of the concrete and the influence of internal and external temperature differences on the sensitive pore radius.

## Figures and Tables

**Figure 1 materials-14-00432-f001:**
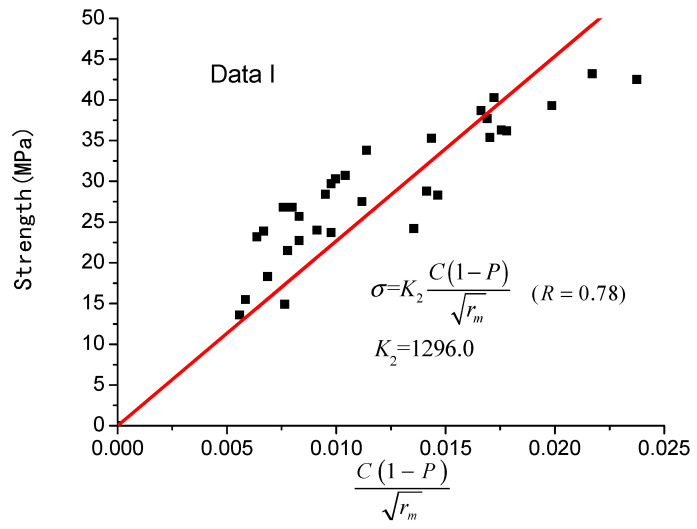
Equation (6) strength model (data I).

**Figure 2 materials-14-00432-f002:**
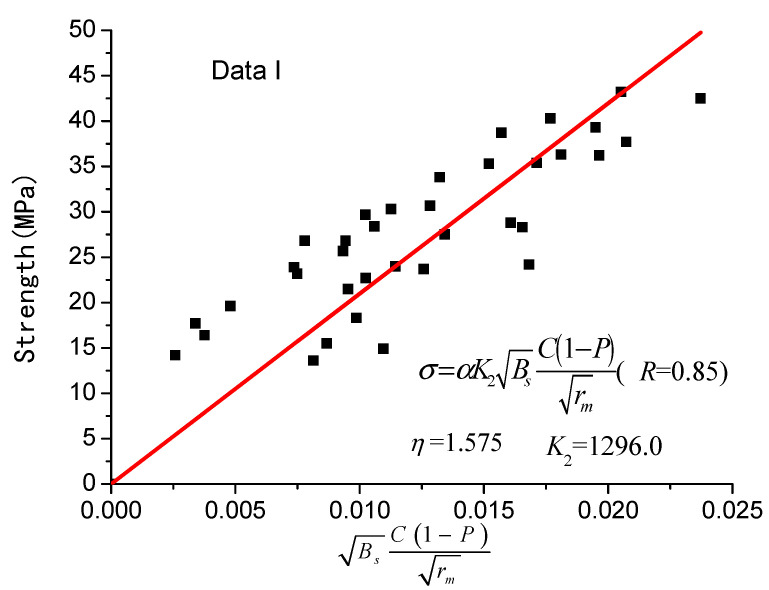
Equation (8) revised strength model (data I).

**Figure 3 materials-14-00432-f003:**
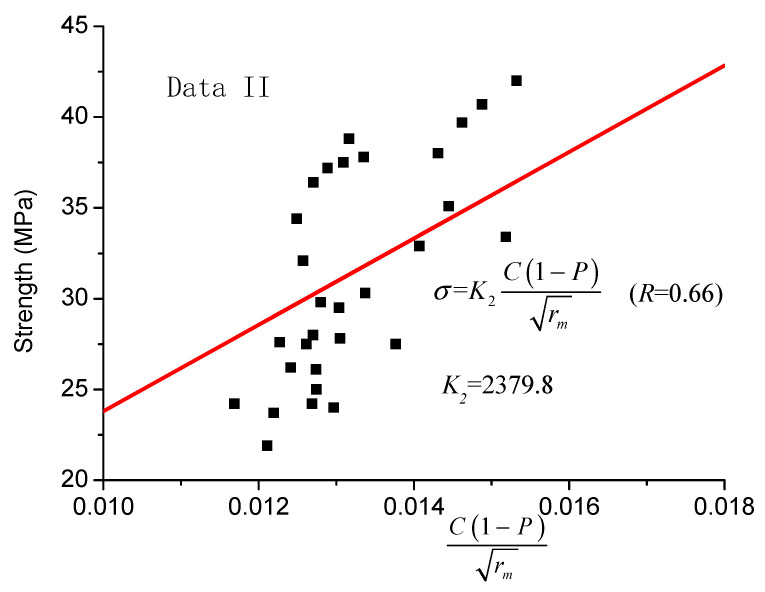
Equation (6) strength model (data II).

**Figure 4 materials-14-00432-f004:**
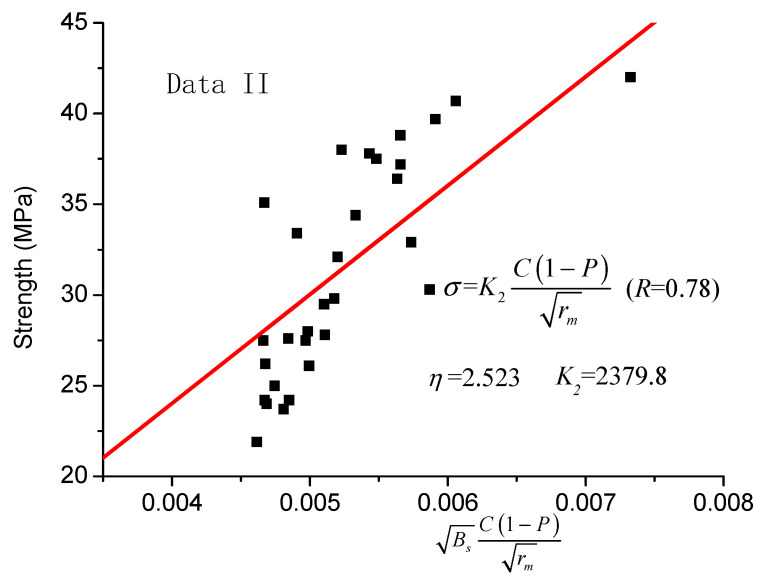
Equation (8) revised strength model (data II).

**Table 1 materials-14-00432-t001:** Concrete parameters (data I).

Strength (MPa)	*C* (%)	*P* (%)	*r_m_* (nm)	Distribution Rate of Pore Radius (%)			
				<10.6 (nm)	10.6–53 (nm)	53–106 (nm)	>106 (nm)
18.3	10.8	12.96	34.3	23.44	37.86	13.92	30.33
28.4	12.6	11.93	38.7	24.04	31.25	7.93	36.78
26.8	12.6	10.87	58.7	19.90	24.77	8.26	47.53
22.7	12.6	11.10	41.3	18.60	34.59	7.35	39.53
21.5	12.6	13.53	42.3	23.26	32.34	7.40	37.00
27.5	12.6	12.75	26.6	26.60	39.69	9.87	23.81
29.7	12.6	10.80	39.3	23.19	32.55	7.56	36.71
26.8	12.6	10.83	52.9	21.44	25.31	7.93	44.94
30.3	13.9	11.80	45.8	15.27	38.72	11.39	34.62
35.3	15.2	11.22	31.2	24.16	39.61	7.25	28.98
40.3	16.9	11.50	30.4	25.06	39.44	7.89	27.61
43.2	19.3	9.26	28.1	24.46	38.59	8.96	27.99
38.7	19.3	10.38	41.9	19.56	34.55	7.19	38.71
28.3	19.3	16.55	34.2	26.13	36.13	6.77	30.97
42.5	19.3	9.50	23.0	28.79	42.44	8.83	19.94
39.3	19.3	9.63	30.3	27.40	37.83	8.82	25.93
15.5	10.8	11.22	41.6	22.37	34.14	6.39	37.10
24.0	12.6	12.04	35.4	23.66	37.70	8.27	30.37
23.2	12.6	11.39	71.3	15.08	27.07	7.64	50.21
14.9	12.6	12.23	31.2	29.78	30.61	7.38	32.20
13.6	12.6	15.37	49.6	21.28	28.91	8.91	40.90
23.7	12.6	12.01	30.5	25.06	37.25	9.31	28.38
25.7	12.6	10.38	47.5	20.52	32.48	7.12	39.88
23.9	12.6	10.40	68.3	15.17	29.05	7.71	48.07
30.7	13.9	11.30	43.0	18.14	38.97	8.82	34.06
33.8	15.2	13.55	45.0	18.15	36.89	6.91	38.07
37.7	16.9	11.85	29.3	23.35	43.48	7.09	26.08
35.4	19.3	9.90	36.9	22.35	35.78	7.16	34.67
28.8	19.3	9.92	43.6	19.34	37.29	6.35	37.02
24.2	19.3	13.31	36.9	22.41	39.29	7.41	30.86
36.2	19.3	9.28	35.0	19.45	44.05	8.20	28.25
36.3	19.3	9.54	35.9	23.79	38.58	5.79	32.46
14.2	12.1	33.7	146.9	11.45	14.14	8.94	65.46
16.4	14.4	33.14	126.7	12.27	16.29	10.53	60.89
17.7	12.1	33.60	122.1	12.89	15.85	11.60	59.67
19.6	14.4	31.70	109.1	14.14	18.11	10.38	57.03

**Table 2 materials-14-00432-t002:** Relation degree of Grey Relation Entropy (GRE) (data I).

Relation Degree	*P*	*r_m_*	*C*	Distribution Rate of Pore Radius (%)			
				<10.6 nm	10.6–53 nm	53–106 nm	>106 nm
*D*(*x_i_*)/%	*D*(*x*_1_)	*D*(*x*_2_)	*D*(*x*_3_)	*D*(*x*_4_)	*D*(*x*_5_)	*D*(*x*_6_)	*D*(*x*_7_)
	99.26	99.17	99.90	99.81	99.91	99.70	99.57

**Table 3 materials-14-00432-t003:** Concrete parameters (data II).

Type	Strength (MPa)	*P*/%	*r_m_*/µm	Distribution Rate of Pore Size (%)					
				10–200 µm	200–400 µm	400–600 µm	600–800 µm	800–1200 µm	1200–1600 µm
0	42	1.05	91	0.1	0.11	0.18	0.24	0.20	0.21
A1	28	5.78	120	0.5	1.55	1.87	0.89	0.65	0.32
A2	27.8	6.07	113	0.56	1.76	1.89	0.93	0.55	0.39
A3	24.2	8.28	114	0.81	2.22	2.82	1.21	0.75	0.46
B1	27.5	6.81	119	0.48	2.00	2.71	0.93	0.47	0.22
B2	26.1	7.09	116	0.63	1.85	2.51	1.09	0.55	0.45
B3	25	8.29	113	0.90	2.38	2.96	1.15	0.59	0.31
C1	39.7	2.51	97	0.44	0.64	0.57	0.41	0.21	0.24
C2	30.3	4.62	111	0.46	0.94	1.44	0.89	0.56	0.32
D1	37.8	2.60	116	0.28	0.62	0.75	0.43	0.28	0.23
D2	29.8	5.87	118	0.31	1.51	2.36	0.96	0.49	0.24
E1	37.2	1.66	127	0.20	0.2	0.50	0.32	0.18	0.17
E2	34.4	2.47	133	0.21	0.23	0.84	0.45	0.44	0.29
E3	27.6	6.74	126	0.57	1.60	2.81	1.05	0.53	0.19
F1	24.2	6.70	139	0.23	1.55	2.86	1.07	0.59	0.40
F2	26.2	7.18	122	0.52	2.07	2.44	1.02	0.63	0.49
F3	23.7	7.33	126	0.55	1.87	2.70	1.14	0.68	0.39
G1	29.5	5.35	115	0.49	1.31	1.79	0.82	0.52	0.41
G2	24.0	8.34	109	0.92	2.41	2.94	1.09	0.57	0.41
G3	21.9	8.68	124	0.82	2.36	3.07	1.26	0.70	0.47
H1	40.7	1.81	95	0.36	0.45	0.41	0.30	0.15	0.14
H2	38.0	2.62	101	0.38	0.61	0.73	0.35	0.35	0.20
H3	32.9	3.31	103	0.37	0.79	1.04	0.55	0.36	0.20
H4	35.1	3.16	98	0.62	0.97	0.90	0.33	0.23	0.10
H5	33.4	4.12	87	0.61	1.39	1.31	0.43	0.25	0.13
H6	27.5	8.21	97	1.06	2.56	2.74	1.07	0.50	0.28
L1	38.8	1.95	121	0.28	0.34	0.50	0.36	0.25	0.24
L2	37.5	2.11	122	0.31	0.39	0.55	0.37	0.31	0.17
L3	36.4	2.29	129	0.32	0.40	0.62	0.45	0.23	0.26
L4	32.1	3.33	129	0.50	0.65	1.20	0.57	0.29	0.12

**Table 4 materials-14-00432-t004:** Relation degree of GRE (data II).

Relation Degree	*P*	*r_m_*	Distribution Rate of Pore Radius (%)					
			10–200 µm	200–400 µm	400–600 µm	600–800 µm	800–1200 µm	1200–1600 µm
*D*(*x_i_*)/%	*D*(*x*_1_)	*D*(*x*_2_)	*D*(*x*_3_)	*D*(*x*_4_)	*D*(*x*_5_)	*D*(*x*_6_)	*D*(*x*_7_)	*D*(*x*_8_)
	99.28	99.74	99.70	99.12	99.60	99.81	99.66	99.29

**Table 5 materials-14-00432-t005:** Regression results of strength formula.

Equation Number	Equation	Regression Equation	R
Equation (6)	$\sigma {=K}_{2}\frac{C\left(1-P\right)}{\sqrt{r_{m}}}$	$\sigma =1296.0\frac{C\left(1-P\right)}{\sqrt{r_{m}}}$	0.780
Equation (8)	$\sigma =\alpha K_{2}\sqrt{B_{s}}\frac{C\left(1-P\right)}{\sqrt{r_{m}}}$	$\sigma =2041.2\sqrt{B_{s}}\frac{C\left(1-P\right)}{\sqrt{r_{m}}}\;$	0.850
Equation (9)	$\sigma =\sigma_{0}{\left(1-p\right)}^{n}$	$\sigma =87.5{\left(1-P\right)}^{9.17}$	0.554
Equation (10)	$\sigma =\sigma_{0}\left(1-AP\right)$	σ=57.3(1−4.3P)	0.522
Equation (11)	$\sigma =\sigma_{0}\exp(-BP)$	$\sigma =95.1e^{\left(-10.5P\right)}$	0.557
Equation (12)	$\sigma =k\ln(\frac{\sigma_{0}}{p})$	$\sigma =31.5\ln(\frac{0.286}{P})$	0.551
Equation (13)	$\sigma =K\frac{\sigma_{0}\left(1-P\right)}{\sqrt{r_{m}}}$	$\sigma =203.1\frac{\left(1-P\right)}{\sqrt{r_{m}}}$	0.578

**Table 6 materials-14-00432-t006:** Test constants and strength values.

*K* _2_	*ƞ*	Test Strength	Equation (6) Strength	Error (%)	Equation (8) Strength	Error (%)
2379.8	2.523	42	36.46	13.19	43.99	4.73
2379.8	2.523	28	30.23	7.97	29.94	6.91
2379.8	2.523	27.8	31.06	11.72	30.68	10.35
2379.8	2.523	24.2	30.19	24.77	29.13	20.36
2379.8	2.523	27.5	30.03	9.19	28.00	1.82
2379.8	2.523	26.1	30.32	16.18	30.00	14.94
2379.8	2.523	25	30.32	21.30	28.50	14.00
2379.8	2.523	39.7	34.79	12.36	35.48	10.62
2379.8	2.523	30.3	31.82	5.02	35.24	16.31
2379.8	2.523	37.8	31.79	15.91	32.62	13.70
2379.8	2.523	29.8	30.46	2.21	31.08	4.30
2379.8	2.523	37.2	30.67	17.55	33.98	8.65
2379.8	2.523	34.4	29.73	13.59	32.02	6.93
2379.8	2.523	27.6	29.20	5.81	29.09	5.38
2379.8	2.523	24.2	27.82	14.94	28.05	15.91
2379.8	2.523	26.2	29.54	12.74	28.09	7.23
2379.8	2.523	23.7	29.02	22.44	28.88	21.84
2379.8	2.523	29.5	31.02	5.16	30.65	3.89
2379.8	2.523	24	30.86	28.58	28.15	17.30
2379.8	2.523	21.9	28.83	31.62	27.71	26.54
2379.8	2.523	40.7	35.41	13.00	36.38	10.62
2379.8	2.523	38	34.06	10.37	31.41	17.34
2379.8	2.523	32.9	33.49	1.79	34.45	4.70
2379.8	2.523	35.1	34.38	2.04	28.04	20.12
2379.8	2.523	33.4	36.13	8.18	29.45	11.81
2379.8	2.523	27.5	32.76	19.12	29.84	8.52
2379.8	2.523	38.8	31.33	19.25	33.97	12.45
2379.8	2.523	37.5	31.15	16.93	32.92	12.22
2379.8	2.523	36.4	30.24	16.93	33.82	7.08
2379.8	2.523	32.1	29.92	6.80	31.23	2.70

where, due to the consideration of the sensitive pore radius in the model, the strength calculated by the revised Equation (8) is closer to the test value.

## Data Availability

No new data were created or analyzed in this study. Data sharing is not applicable to this article.

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
