# Peer review of "Study of Concrete Strength and Pore Structure Model Based on Grey Relation Entropy"

_materials, 2021, doi:10.3390/ma14020432_

Round 1

Reviewer 1 Report

The articles present the issues of the GRE theory. A model for determining the compressive strength of concrete as a function of the pore size - the sensitivity factor was proposed. Below are my comments:

  1. Chapter 1: Knowledge Analysis contains many publications that are not very up-to-date. There is no clearly formulated goal and scope of work,
  2. Chapter 2.1: The GRE theory should be described in more detail - the origins of formulas (1) - (3) should be given.
  3. Chapter 2.2: the mercury intrusion test method should be described - including possible reliability of the results.
  4. Chapter 3: As in chapter 2.1, the cited formulas (4) - (8) should be better discussed, at least the assumptions and application ranges should be given.
  5. Chapter 4: The quoted formulas (9) - (13) should be discussed in the same way as in p. 2 and 4. I believe that the method is not applicable. The analyzes were performed only in a certain strength range. Therefore, the modified equations should take this into account.
  6. Chapter 5. The conclusions are quite modest, giving the impression that the topic has been exhausted. Do the authors plan to perform their own porosity tests of other concretes?

Author Response

Dear Editor,

Thank you very much for your letter and the comments from the referees about our manuscript, Study of concrete strength and pore structure model based on Grey Relation Entropy. We acknowledge your comments and constructive suggestions very much, which are valuable for improving the quality of our manuscript. We have provided itemized responses in blue colored text. And we have checked and revised the manuscript according to the reviewers’ comments in detail. All changes in the revised manuscript were highlighted using the yellow background for easy checking and editing. We hope, with these modifications and improvements based on your suggestions and the reviewers’ comments, the quality of our manuscript would meet the publication standard of Current Science. We submit here the revised manuscript as well as the response to reviewer comments of our manuscript. If you have any question about this paper, please don’t hesitate to let me know.

Once again, thank you very much for your help to our paper processing.

Sincerely  yours,

Min Zhang

School of Civil Engineering and Communication, North China University of Water Resources and Electric Power

Junfeng Guan

School of Civil Engineering and Communication, North China University of Water Resources and Electric Power

Response to reviewer comments point by point

Reviewer #1:

The articles present the issues of the GRE theory. A model for determining the compressive strength of concrete as a function of the pore size - the sensitivity factor was proposed. Below are my comments:

1.Chapter 1: Knowledge Analysis contains many publications that are not very up-to-date. There is no clearly formulated goal and scope of work.

Many thanks.

Some references have been updated and added.

“ .…….

[1]Y . Guo , S .Wu , Z. Lyu , et al. Pore structure characteristics and performance of construction waste composite powder-modified concrete[J]. Construction and Building Materials, 2020:121262.

[2] H. Liu , C. Liu , G. Bai, et al. Impact of chloride intrusion on the pore structure of recycled aggregate concrete based on the recycled aggregate porous interface[J]. Construction and Building Materials, 2020, 259:120397.

.…….

[7]E. Ryshkewitch, Compression strength of porous sintered alumina and zirconia, J. Am. Ceram. Soc. 36 (2) (2010) 65–68.

[8]S. Zhang , K. Cao , C. Wang , et al. Influence of the porosity and pore size on the compressive and splitting strengths of cellular concrete with millimeter-size pores[J]. Construction and Building Materials, 2019, 235.M.  

.…….

[23] M. Shilpa , G. S. Prakash , M. R. Shivakumar. A combinatorial approach to optimize the properties of green sand used in casting mould[J]. Materials today: proceedings, 2020.

[24]R. Guo , T. Nian , F. Zhou. Analysis of factors that influence anti-rutting performance of asphalt pavement[J]. Construction and Building Materials, 2020, 254:119237.

.…….

[29]H. Wu, P. Li , T. Nian , et al. Evaluation of asphalt and asphalt mixtures' water stability method under multiple freeze-thaw cycles[J]. Construction and Building Materials, 2019, 228(Dec.20):117089.1-117089.15.

[30]N. Sanghvi, D. Vora, J. Patel et al., Optimization of end milling of Inconel 825 with coated tool: A mathematical comparison between GRA, TOPSIS and Fuzzy Logic methods, Materials Today: Proceedings, https://doi.org/10.1016/j.matpr.2020.06.413.

[31]E. Berodier , J. Bizzozero , A. Muller . Mercury Intrusion Porosimetry[M]. A Practical Guide to Microstructural Analysis of Cementitious Materials. 2015..…….”

  1. Chapter 2.1: The GRE theory should be described in more detail - the origins of formulas (1) - (3) should be given.

Many thanks for the nice suggestion.

The GRE theory has been described in more detail and the origins of formulas (1) - (3) has been given as follows.

“ .…….

2.1 GRE theory

    In order to solve the problem of partial correlation tendency and loss of individual information, which cause significant deviation of the gray correlation degree and affects the analysis results, based on gray correlation analysis, GRE theory is introduced to quantitatively describe the relation degree between various factors. The calculation theory is as follows [29-30].

    The reference sequence is X0(k) and the comparative data column is Xi(k)(k=1,2,…m; i=1,2,…n)  .…….”

  1. Chapter 2.2: the mercury intrusion test method should be described - including possible reliability of the results.

Many thanks.

The mercury intrusion test method has been described - including possible reliability of the results.

“ .…….  The GRE method is used to select the most sensitive factors that affect the strength of concrete, in order to find the relationship between them.

The mercury intrusion porosimetry (MIP) method is used to test the pore characteristics of concrete materials, which can well detect the porosity, pore size distribution and other pore properties[31]. Hence, the GRE analysis is performed on the concrete mercury intrusion test data, denoted as data I  .…….”

  1. Chapter 3: As in chapter 3.1, the cited formulas (4) - (8) should be better discussed, at least the assumptions and application ranges should be given.

Many thanks for the nice suggestion..

The question has been explained in the revised manuscript, the following content has been added:

“ .……. Replacing ƞ and α into Eq.(4),Eq.(5) is formed, in which it is noted that the sensitivity coefficient ƞ is proportional to the arithmetic square root of the sensitive pore size distribution rate Bs.

                            (5)

To be noted , the sensitive pore size is the pore size corresponding to the factor with the largest sensitivity coefficient which is obtained through GRE analysis. Thus, the sensitive pore size has the greatest influence on the strength of the concrete. Eq.(4) and Eq.(5) are formulas based on certain assumptions that each pore size is an independent circle, in a certain section of concrete.

.…….

Based on Eq. (8),it is noted that the compressive strength of concrete can be determined by the cement content, porosity, average pore size and sensitive pore size distribution rate.

To be noted , Eq.(8) proposed in this paper firstly performs GRE analysis to find the sensitive pore size, which leads to certain limitations in the use of Eq.(8). The use of the Eq.(8) requires that the pore structure data must meet conditions that the pore size range is consistent and the concrete strength grade and age should also be consistent. .…….”

  1. Chapter 4: The quoted formulas (9) - (13) should be discussed in the same way as in p. 2 and 4. I believe that the method is not applicable. The analyzes were performed only in a certain strength range. Therefore, the modified equations should take this into account.

Many thanks.

The question has been explained in the revised manuscript:

 “ .……. .…….

Based on Eq. (8),it is noted that the compressive strength of concrete can be determined by the cement content, porosity, average pore size and sensitive pore size distribution rate.

To be noted , Eq.(8) proposed in this paper firstly performs GRE analysis to find the sensitive pore size, which leads to certain limitations in the use of Eq.(8). The use of the Eq.(8) requires that the pore structure data must meet conditions that the pore size range is consistent and the concrete strength grade and age should also be consistent. .…….”

  1. Chapter 5. The conclusions are quite modest, giving the impression that the topic has been exhausted. Do the authors plan to perform their own porosity tests of other concretes?

Many thanks for the suggestion.

The question has been explained in the conclusion of the revised manuscript, the following content has been added:

 “ .…….

(4)In the revised model, the concrete's strength is related not only to porosity, average pore size and cement content, but also to sensitive pore size. The consideration of the influence of the sensitive pore size on the strength of concrete, leads to better estimation of the concrete's strength.

(5)This paper only considers the influence of sensitive pore size on the strength of concrete, but does not consider the influence of capillary pores, gel pores, location of pores and interconnectivity of pores on concrete  strength. In the future, the author intends to conduct a porosity test on studied concrete, taking into account the influence of the pore structure on the permeability of the concrete and the influence of internal and external temperature differences on the sensitive pore size. .…….”

Reviewer 2 Report

The paper is interesting, but it exposes the arguments in an often too synthetic way, to the detriment of the understanding. Authors should consider that not all readers are familiar with the topics covered and should provide all useful information for a full understanding of the text. It is recommended that the text be revised in the light of these considerations.
Some other minor and major recommendations are listed below.

Line 21
“average pore size”
“Size” is too general a term. You should define what parameter you chose as representative of the average size. Based on the comment on line 142 (see below) it appears that this is a radius.

Line 24
Here you define the distribution rate of sensitive pore size, but you should first define what a sensitive pore is.

Line 26

“average pore diameter of sensitive pores”
See the comment on line 143: it appears that rm is a radius, not a diameter.

Line 58
Replace the comma before the word “Facing” with a full stop.

Line 65
“However, this formula does not consider pore size distribution et al. factors that influence the concrete strength.”
You probably mean “all factors”

Line 77
As noted for the nomenclature section, you speak of “sensitive pore” without having defined what you mean by sensitive pore. Provide a definition of sensitive pore the first time you use this term.

Line 142
“the average pore diameter is rm”
Based on the formula of A0, given in line 145, it appears that rm is the average pore radius, not the diameter.

Line 143
“the average pore diameter of sensitive pores is rms”
Same observation as for line 142: according to the formula of As, this appears to be the average pore radius.

Author Response

Dear Editor,

Thank you very much for your letter and the comments from the referees about our manuscript, Study of concrete strength and pore structure model based on Grey Relation Entropy. We acknowledge your comments and constructive suggestions very much, which are valuable for improving the quality of our manuscript. We have provided itemized responses in blue colored text. And we have checked and revised the manuscript according to the reviewers’ comments in detail. All changes in the revised manuscript were highlighted using the yellow background for easy checking and editing. We hope, with these modifications and improvements based on your suggestions and the reviewers’ comments, the quality of our manuscript would meet the publication standard of Current Science. We submit here the revised manuscript as well as the response to reviewer comments of our manuscript. If you have any question about this paper, please don’t hesitate to let me know.

Once again, thank you very much for your help to our paper processing.

Sincerely  yours,

Min Zhang

School of Civil Engineering and Communication, North China University of Water Resources and Electric Power

Junfeng Guan

School of Civil Engineering and Communication, North China University of Water Resources and Electric Power

Response to reviewer comments point by point

Reviewer #2:

The paper is interesting.

Many thanks for the positive comments and support.

But it exposes the arguments in an often too synthetic way, to the detriment of the understanding. Authors should consider that not all readers are familiar with the topics covered and should provide all useful information for a full understanding of the text. It is recommended that the text be revised in the light of these considerations. Some other minor and major recommendations are listed below.

  1. Line 21
    “average pore size”
    “Size” is too general a term. You should define what parameter you chose as representative of the average size. Based on the comment on line 142 (see below) it appears that this is a radius.

Many thanks for the suggestion.

Because of average pore size=k*total pore volume/specific surface area, average pore size refers to radius.

  1. Line 24
    Here you define the distribution rate of sensitive pore size, but you should first define what a sensitive pore is.
    Many thanks for the reasonable suggestion.

The sensitive pore size has been defined in the revised manuscript:

“ .……. The values of relation degree of GRE, calculated by Eq. (3) and list in Table 2 are close, following the descending order: D(x5)> D(x3)> D(x6) > D(x4)> D(x7)> D(x1)> D(x2).the sensitive pore size is the pore size corresponding to the factor with the largest sensitivity coefficient. The maximum value of the relation degree is D(x5), thus, …….”

  1. Line 26
    “average pore diameter of sensitive pores”
    See the comment on line 143: it appears that rm is a radius, not a diameter.
    Many thanks for the suggestion.

Because of average pore size=k*total pore volume/specific surface area, average pore size refers to radius.

The question has been modified in the revised manuscript:

“ .…….rms                       average pore size of sensitive pores .…….”

  1. Line 58
    Replace the comma before the word “Facing” with a full stop.
    Many thanks.

The question has been corrected in the revised manuscript.

“ .……. structure. Facing the problem of .…….”

  1. Line 65
    “However, this formula does not consider pore size distribution et al. factors that influence the concrete strength.”You probably mean “all factors”

Many thanks.

 “However, this formula does not consider pore size distribution factors that influence the concrete strength.”  

"et al." has been removed.

  1. Line 77
    As noted for the nomenclature section, you speak of “sensitive pore” without having defined what you mean by sensitive pore. Provide a definition of sensitive pore the first time you use this term.
    Many thanks.

The question has been explained in the revised manuscript:

 “ .……. Replacing ƞ and α into Eq.(4),Eq.(5) is formed, in which it is noted that the sensitivity coefficient ƞ is proportional to the arithmetic square root of the sensitive pore size distribution rate Bs.

                            (5)

To be noted , the sensitive pore size is the pore size corresponding to the factor with the largest sensitivity coefficient which is obtained through GRE analysis. Thus, the sensitive pore size has the greatest influence on the strength of the concrete. Eq.(4) and Eq.(5) are formulas based on certain assumptions that each pore size is an independent circle, in a certain section of concrete. …….”

  1. Line 142
    “the average pore diameter is rm
    Based on the formula of A0, given in line 145, it appears that rm is the average pore radius, not the diameter.
    Many thanks for the suggestion.

Because of average pore size=k*total pore volume/specific surface area, average pore size refers to radius.

The question has been corrected in the revised manuscript:

    “ .……. the average pore size is rm.…….”

  1. Line 143
    “the average pore diameter of sensitive pores is rms”. Same observation as for line 142: according to the formula of As, this appears to be the average pore radius.

Many thanks for the suggestion.

Because of average pore size=k*total pore volume/specific surface area, average pore size refers to radius.

The question has been corrected in the revised manuscript:

    “ .……. the average pore size is rms.…….”

Reviewer 3 Report

Reviewers' comments:

Manuscript ID: materials-1051319

Title: Study of concrete strength and pore structure model based on Grey Relation Entropy.

Manuscript Type: Article.

Reviewers' comments:

The manuscript describes the Study of concrete strength and pore structure model based on Grey Relation Entropy. The manuscript needs a detailed editing. Some markings are made to just illustrate the extent of editing needed. A thorough revision addressing all the concerns is needed and if the authors are prepared to do that it can be considered for a review of the revised manuscript.

The authors need to consider the following comments

- Sensitive pore size not explained by the authors clearly and what is the limitation of pore size?

- Is it sensitive pore size means the pore will be closed later? I f so how this parameter can be considered as main for compressive strength of concrete?

- Whether these sensitive pore size are interconnected. How it can be ensured. If so, permeability of the concrete should also be considered.

- The proposed modified equation gives only the relationship between pore size and strength but other parameters like degree of hydration, w/c ratio also influences the strength of concrete.

- Author should mention the limitation of proposed equation like grade of concrete, admixture to be used and age of concrete etc.,

- For the proposed equation what are the assumptions made? What are the conditions in which the equation can satisfy?

- K values given are varying and authors should mention the parameters which are influencing the K value in the equation.

- Distribution of capillary pores, gel pores also should be considered for getting strength.

- Location of pores and interconnectivity of pores leads to transport movement of liquid inside concrete which may affect the performance of concrete. Author should address these points.

- Temperature of concrete inside and outside may alter the sensitive pore size and author should consider this parameter also into account.

- References: author should use order and there are recent references in 2019 and 2020 treating the same subject, you can use.

- Several faults: are added or missing spaces between words: see manuscript file.

Author Response

Dear Editor,

Thank you very much for your letter and the comments from the referees about our manuscript, Study of concrete strength and pore structure model based on Grey Relation Entropy. We acknowledge your comments and constructive suggestions very much, which are valuable for improving the quality of our manuscript. We have provided itemized responses in blue colored text. And we have checked and revised the manuscript according to the reviewers’ comments in detail. All changes in the revised manuscript were highlighted using the yellow background for easy checking and editing. We hope, with these modifications and improvements based on your suggestions and the reviewers’ comments, the quality of our manuscript would meet the publication standard of Current Science. We submit here the revised manuscript as well as the response to reviewer comments of our manuscript. If you have any question about this paper, please don’t hesitate to let me know.

Once again, thank you very much for your help to our paper processing.

Sincerely  yours,

Min Zhang

School of Civil Engineering and Communication, North China University of Water Resources and Electric Power

Junfeng Guan

School of Civil Engineering and Communication, North China University of Water Resources and Electric Power

Response to reviewer comments point by point

Reviewer #3:

The manuscript describes the Study of concrete strength and pore structure model based on Grey Relation Entropy. The manuscript needs a detailed editing. Some markings are made to just illustrate the extent of editing needed. A thorough revision addressing all the concerns is needed and if the authors are prepared to do that it can be considered for a review of the revised manuscript.

The authors need to consider the following comments.

  1. Sensitive pore size not explained by the authors clearly and what is the limitation of pore size?

Many thanks for the nice suggestion.

The question has been explained on page 9 of the revised manuscript, the following content has been added:

 “ .……. Replacing ƞ and α into Eq.(4),Eq.(5) is formed, in which it is noted that the sensitivity coefficient ƞ is proportional to the arithmetic square root of the sensitive pore size distribution rate Bs.

                            (5)

To be noted , the sensitive pore size is the pore size corresponding to the factor with the largest sensitivity coefficient which is obtained through GRE analysis. Thus, the sensitive pore size has the greatest influence on the strength of the concrete. Eq.(4) and Eq.(5) are formulas based on certain assumptions that each pore size is an independent circle, in a certain section of concrete.

.……. ”

  1. Is it sensitive pore size means the pore will be closed later? If so how this parameter can be considered as main for compressive strength of concrete?

Many thanks.

The question has been explained, as follows:

The revised equation proposed in the paper is based on the GRE theory, and does not consider whether the pore closed or not. The sensitive pore size is that the pore size corresponding to the factor with the largest sensitivity coefficient which is obtained through GRE analysis, firstly performs GRE analysis to find the sensitive pore size.

  1. Whether these sensitive pore size are interconnected. How it can be ensured. If so, permeability of the concrete should also be considered.

Many thanks.

The question has been explained:

The sensitive pore size are not related to each other. The premise of GRE analysis to determine the sensitive pore size is that the pore size distribution range must be consistent, and the concrete strength grade and age should also be consistent. Considering the influence of concrete permeability on concrete strength and pore structure is a good idea for future research.

In the conclusion of the revised manuscript, the following content has been added:

“ .…….

(4)In the revised model, the concrete's strength is related not only to porosity, average pore size and cement content, but also to sensitive pore size. The consideration of the influence of the sensitive pore size on the strength of concrete, leads to better estimation of the concrete's strength.

(5)This paper only considers the influence of sensitive pore size on the strength of concrete, but does not consider the influence of capillary pores, gel pores, location of pores and interconnectivity of pores on concrete  strength. In the future, the author intends to conduct a porosity test on studied concrete, taking into account the influence of the pore structure on the permeability of the concrete and the influence of internal and external temperature differences on the sensitive pore size. .…….”

    4.The proposed modified equation gives only the relationship between pore size and strength but other parameters like degree of hydration, w/c ratio also influences the strength of concrete.

Many thanks for the suggestion.

The question has been explained, as follows:

The concrete's strength is related not only to porosity and average pore size, but also to water-cement ratio and the hydration degree of the cement. The revised model proposed in this paper takes into account the influence of cement content and sensitive pore size on the strength of concrete.

  1. Author should mention the limitation of proposed equation like grade of concrete, admixture to be used and age of concrete etc. For the proposed equation what are the assumptions made? What are the conditions in which the equation can satisfy?

Many thanks.

The question has been explained in the revised manuscript, the following content has been added:

“ .……. Replacing ƞ and α into Eq.(4),Eq.(5) is formed, in which it is noted that the sensitivity coefficient ƞ is proportional to the arithmetic square root of the sensitive pore size distribution rate Bs.

                            (5)

To be noted , the sensitive pore size is the pore size corresponding to the factor with the largest sensitivity coefficient which is obtained through GRE analysis. Thus, the sensitive pore size has the greatest influence on the strength of the concrete. Eq.(4) and Eq.(5) are formulas based on certain assumptions that each pore size is an independent circle, in a certain section of concrete.

.…….

Based on Eq. (8),it is noted that the compressive strength of concrete can be determined by the cement content, porosity, average pore size and sensitive pore size distribution rate.

To be noted , Eq.(8) proposed in this paper firstly performs GRE analysis to find the sensitive pore size, which leads to certain limitations in the use of Eq.(8). The use of the Eq.(8) requires that the pore structure data must meet conditions that the pore size range is consistent and the concrete strength grade and age should also be consistent.

.…….”

  1. K values given are varying and authors should mention the parameters which are influencing the K value in the equation.

Many thanks.

The question has been explained in the revised manuscript, the following content has been added:

“ .……. introducing also the cement content parameter,  denoted as C and a test constant K2, which can be obtained by fitting test data, as shown in Eq. (6).

                         (6)

where, K2 is a test constant, which is mainly related to the strength of the concrete matrix, and is related to factors such as the type of aggregate, the type of cement and the cement content.

    In this paper, the influence of the sensitive pore size is taken into account, introducing the sensitivity coefficient.  …….”

  1. Distribution of capillary pores, gel pores also should be considered for getting strength. Location of pores and interconnectivity of pores leads to transport movement of liquid inside concrete which may affect the performance of concrete. Author should address these points. Temperature of concrete inside and outside may alter the sensitive pore size and author should consider this parameter also into account.

Many thanks. Provides some good ideas for further research.

The question has been explained in the conclusion of the revised manuscript, the following content has been added:

“ .…….

(4)In the revised model, the concrete's strength is related not only to porosity, average pore size and cement content, but also to sensitive pore size. The consideration of the influence of the sensitive pore size on the strength of concrete, leads to better estimation of the concrete's strength.

(5)This paper only considers the influence of sensitive pore size on the strength of concrete, but does not consider the influence of capillary pores, gel pores, location of pores and interconnectivity of pores on concrete  strength. In the future, the author intends to conduct a porosity test on studied concrete, taking into account the influence of the pore structure on the permeability of the concrete and the influence of internal and external temperature differences on the sensitive pore size.  .…….”

  1. References: author should use order and there are recent references in 2019 and 2020 treating the same subject, you can use.

Many thanks.

The references has been updated in the revised manuscript.

“ .…….

[1]Y . Guo , S .Wu , Z. Lyu , et al. Pore structure characteristics and performance of construction waste composite powder-modified concrete[J]. Construction and Building Materials, 2020:121262.

[2] H. Liu , C. Liu , G. Bai, et al. Impact of chloride intrusion on the pore structure of recycled aggregate concrete based on the recycled aggregate porous interface[J]. Construction and Building Materials, 2020, 259:120397.

.…….

[8]S. Zhang , K. Cao , C. Wang , et al. Influence of the porosity and pore size on the compressive and splitting strengths of cellular concrete with millimeter-size pores[J]. Construction and Building Materials, 2019, 235.M.  

.…….

[23] M. Shilpa , G. S. Prakash , M. R. Shivakumar. A combinatorial approach to optimize the properties of green sand used in casting mould[J]. Materials today: proceedings, 2020.

[24]R. Guo , T. Nian , F. Zhou. Analysis of factors that influence anti-rutting performance of asphalt pavement[J]. Construction and Building Materials, 2020, 254:119237.

.…….

[29]H. Wu, P. Li , T. Nian , et al. Evaluation of asphalt and asphalt mixtures' water stability method under multiple freeze-thaw cycles[J]. Construction and Building Materials, 2019, 228(Dec.20):117089.1-117089.15.

[30]N. Sanghvi, D. Vora, J. Patel et al., Optimization of end milling of Inconel 825 with coated tool: A mathematical comparison between GRA, TOPSIS and Fuzzy Logic methods, Materials Today: Proceedings, https://doi.org/10.1016/j.matpr.2020.06.413.

.…….”

  1. Several faults: are added or missing spaces between words: see manuscript file.

Many thanks.

The question has been modified in the revised manuscript.

Round 2

Reviewer 1 Report

Many thanks to the authors for responding to my comments. I believe my doubts have been resolved. The corrections made to the article are sufficient. However, I believe that the list of designations at the beginning of the article can be omitted. Lists of all variables should appear next to each formula. It is imperative to correct the numbers in the arrays because they go to the second line.

Author Response

Dear Editor,

Thank you very much for your letter and the comments from the referees about our manuscript, Study of concrete strength and pore structure model based on Grey Relation Entropy. We acknowledge your comments and constructive suggestions very much, which are valuable for improving the quality of our manuscript. We have provided itemized responses in blue colored text. And we have checked and revised the manuscript according to the reviewers’ comments in detail. All changes in the revised manuscript were highlighted using the yellow background for easy checking and editing. We hope, with these modifications and improvements based on your suggestions and the reviewers’ comments, the quality of our manuscript would meet the publication standard of Current Science. We submit here the revised manuscript as well as the response to reviewer comments of our manuscript. If you have any question about this paper, please don’t hesitate to let me know.

Once again, thank you very much for your help to our paper processing.

Sincerely  yours,

Junfeng Guan

School of Civil Engineering and Communication, North China University of Water Resources and Electric Power

Response to reviewer comments point by point

Reviewer #1:

1.Many thanks to the authors for responding to my comments. I believe my doubts have been resolved. The corrections made to the article are sufficient. However, I believe that the list of designations at the beginning of the article can be omitted. Lists of all variables should appear next to each formula. It is imperative to correct the numbers in the arrays because they go to the second line.

Many thanks for the nice suggestion.

The list of designations at the beginning of the article has been removed, all variables in the list appear next to each formula.

The numbers in the arrays have been corrected, as followed.

Table 1. Concrete parameters (data I).

Strength

(MPa)

C

(%)

P

(%)

rm

(nm)

Distribution rate of pore radius (%)

<10.6

(nm)

10.6-53

( nm)

53-106

( nm)

>106 (nm)

18.3

10.8

12.96

34.3

23.44

37.86

13.92

30.33

28.4

12.6

11.93

38.7

24.04

31.25

7.93

36.78

26.8

12.6

10.87

58.7

19.90

24.77

8.26

47.53

22.7

12.6

11.10

41.3

18.60

34.59

7.35

39.53

21.5

12.6

13.53

42.3

23.26

32.34

7.40

37.00

27.5

12.6

12.75

26.6

26.60

39.69

9.87

23.81

29.7

12.6

10.80

39.3

23.19

32.55

7.56

36.71

26.8

12.6

10.83

52.9

21.44

25.31

7.93

44.94

30.3

13.9

11.80

45.8

15.27

38.72

11.39

34.62

35.3

15.2

11.22

31.2

24.16

39.61

7.25

28.98

40.3

16.9

11.50

30.4

25.06

39.44

7.89

27.61

43.2

19.3

9.26

28.1

24.46

38.59

8.96

27.99

38.7

19.3

10.38

41.9

19.56

34.55

7.19

38.71

28.3

19.3

16.55

34.2

26.13

36.13

6.77

30.97

42.5

19.3

9.50

23.0

28.79

42.44

8.83

19.94

39.3

19.3

9.63

30.3

27.40

37.83

8.82

25.93

15.5

10.8

11.22

41.6

22.37

34.14

6.39

37.10

24.0

12.6

12.04

35.4

23.66

37.70

8.27

30.37

23.2

12.6

11.39

71.3

15.08

27.07

7.64

50.21

14.9

12.6

12.23

31.2

29.78

30.61

7.38

32.20

13.6

12.6

15.37

49.6

21.28

28.91

8.91

40.90

23.7

12.6

12.01

30.5

25.06

37.25

9.31

28.38

25.7

12.6

10.38

47.5

20.52

32.48

7.12

39.88

23.9

12.6

10.40

68.3

15.17

29.05

7.71

48.07

30.7

13.9

11.30

43.0

18.14

38.97

8.82

34.06

33.8

15.2

13.55

45.0

18.15

36.89

6.91

38.07

37.7

16.9

11.85

29.3

23.35

43.48

7.09

26.08

35.4

19.3

9.90

36.9

22.35

35.78

7.16

34.67

28.8

19.3

9.92

43.6

19.34

37.29

6.35

37.02

24.2

19.3

13.31

36.9

22.41

39.29

7.41

30.86

36.2

19.3

9.28

35.0

19.45

44.05

8.20

28.25

36.3

19.3

9.54

35.9

23.79

38.58

5.79

32.46

14.2

12.1

33.7

146.9

11.45

14.14

8.94

65.46

16.4

14.4

33.14

126.7

12.27

16.29

10.53

60.89

17.7

12.1

33.60

122.1

12.89

15.85

11.60

59.67

19.6

14.4

31.70

109.1

14.14

18.11

10.38

57.03

Table 2 Relation degree of GRE (data I).

Relation degree

P

rm

C

Distribution rate of pore radius (%)

<10.6 nm

10.6-53 nm

53-106 nm

>106 nm

D(xi) / %

D(x1)

D(x2)

D(x3)

D(x4)

D(x5)

D(x6)

D(x7)

99.26

99.17

99.90

99.81

99.91

99.70

99.57

Table 3 Concrete parameters (data II).

Type

Strength

(MPa)

P

/ %

rm

/ µm

Distribution rate of pore size (%)

10-200

µm

200-400

µm

400-600

µm

600-800

µm

800-1200

µm

1200-1600

µm

0

42

1.05

91

0.1

0.11

0.18

0.24

0.20

0.21

A1

28

5.78

120

0.5

1.55

1.87

0.89

0.65

0.32

A2

27.8

6.07

113

0.56

1.76

1.89

0.93

0.55

0.39

A3

24.2

8.28

114

0.81

2.22

2.82

1.21

0.75

0.46

B1

27.5

6.81

119

0.48

2.00

2.71

0.93

0.47

0.22

B2

26.1

7.09

116

0.63

1.85

2.51

1.09

0.55

0.45

B3

25

8.29

113

0.90

2.38

2.96

1.15

0.59

0.31

C1

39.7

2.51

97

0.44

0.64

0.57

0.41

0.21

0.24

C2

30.3

4.62

111

0.46

0.94

1.44

0.89

0.56

0.32

D1

37.8

2.60

116

0.28

0.62

0.75

0.43

0.28

0.23

D2

29.8

5.87

118

0.31

1.51

2.36

0.96

0.49

0.24

E1

37.2

1.66

127

0.20

0.2

0.50

0.32

0.18

0.17

E2

34.4

2.47

133

0.21

0.23

0.84

0.45

0.44

0.29

E3

27.6

6.74

126

0.57

1.60

2.81

1.05

0.53

0.19

F1

24.2

6.70

139

0.23

1.55

2.86

1.07

0.59

0.40

F2

26.2

7.18

122

0.52

2.07

2.44

1.02

0.63

0.49

F3

23.7

7.33

126

0.55

1.87

2.70

1.14

0.68

0.39

G1

29.5

5.35

115

0.49

1.31

1.79

0.82

0.52

0.41

G2

24.0

8.34

109

0.92

2.41

2.94

1.09

0.57

0.41

G3

21.9

8.68

124

0.82

2.36

3.07

1.26

0.70

0.47

H1

40.7

1.81

95

0.36

0.45

0.41

0.30

0.15

0.14

H2

38.0

2.62

101

0.38

0.61

0.73

0.35

0.35

0.20

H3

32.9

3.31

103

0.37

0.79

1.04

0.55

0.36

0.20

H4

35.1

3.16

98

0.62

0.97

0.90

0.33

0.23

0.10

H5

33.4

4.12

87

0.61

1.39

1.31

0.43

0.25

0.13

H6

27.5

8.21

97

1.06

2.56

2.74

1.07

0.50

0.28

L1

38.8

1.95

121

0.28

0.34

0.50

0.36

0.25

0.24

L2

37.5

2.11

122

0.31

0.39

0.55

0.37

0.31

0.17

L3

36.4

2.29

129

0.32

0.40

0.62

0.45

0.23

0.26

L4

32.1

3.33

129

0.50

0.65

1.20

0.57

0.29

0.12

Table 4 Relation degree of GRE (data II).

Relation degree

P

rm

Distribution rate of pore radius (%)

10-200

µm

200-400

µm

400-600

µm

600-800

µm

800-1200

µm

1200-1600

µm

D(xi) / %

D(x1)

D(x2)

D(x3)

D(x4)

D(x5)

D(x6)

D(x7)

D(x8)

99.28

99.74

99.70

99.12

99.60

99.81

99.66

99.29

Table 5 Regression results of strength formula.

Equation number

Equation

Regression equation

R

Eq. (6)

0.780

Eq. (8)

0.850

Eq. (9)

0.554

Eq. (10)

0.522

Eq. (11)

0.557

Eq. (12)

0.551

 Eq. (13)

0.578

Table 6 Test constants and strength values.

K2

ƞ

Test strength

Eq.(6)

strength

Error (%)

Eq.(8)

strength

Error (%)

2379.8

2.523

42

36.46

13.19

43.99

4.73

2379.8

2.523

28

30.23

7.97

29.94

6.91

2379.8

2.523

27.8

31.06

11.72

30.68

10.35

2379.8

2.523

24.2

30.19

24.77

29.13

20.36

2379.8

2.523

27.5

30.03

9.19

28.00

1.82

2379.8

2.523

26.1

30.32

16.18

30.00

14.94

2379.8

2.523

25

30.32

21.30

28.50

14.00

2379.8

2.523

39.7

34.79

12.36

35.48

10.62

2379.8

2.523

30.3

31.82

5.02

35.24

16.31

2379.8

2.523

37.8

31.79

15.91

32.62

13.70

2379.8

2.523

29.8

30.46

2.21

31.08

4.30

2379.8

2.523

37.2

30.67

17.55

33.98

8.65

2379.8

2.523

34.4

29.73

13.59

32.02

6.93

2379.8

2.523

27.6

29.20

5.81

29.09

5.38

2379.8

2.523

24.2

27.82

14.94

28.05

15.91

2379.8

2.523

26.2

29.54

12.74

28.09

7.23

2379.8

2.523

23.7

29.02

22.44

28.88

21.84

2379.8

2.523

29.5

31.02

5.16

30.65

3.89

2379.8

2.523

24

30.86

28.58

28.15

17.30

2379.8

2.523

21.9

28.83

31.62

27.71

26.54

2379.8

2.523

40.7

35.41

13.00

36.38

10.62

2379.8

2.523

38

34.06

10.37

31.41

17.34

2379.8

2.523

32.9

33.49

1.79

34.45

4.70

2379.8

2.523

35.1

34.38

2.04

28.04

20.12

2379.8

2.523

33.4

36.13

8.18

29.45

11.81

2379.8

2.523

27.5

32.76

19.12

29.84

8.52

2379.8

2.523

38.8

31.33

19.25

33.97

12.45

2379.8

2.523

37.5

31.15

16.93

32.92

12.22

2379.8

2.523

36.4

30.24

16.93

33.82

7.08

2379.8

2.523

32.1

29.92

6.80

31.23

2.70

Reviewer 2 Report

The reviewer appreciates the effort made by the author to improve the text of the paper. However, some aspects still deserve further investigation.

Line 19

“average pore size”

The reviewer appreciates the explanation of the authors based on the formula of the average pore size and the correction made to the term “diameter”. However, the reviewer remains of the opinion that the term “size” is too general, as the characteristic size of an object assimilated to a sphere is not necessarily a radius. If it is a radius, it is good to call it “average pore radius”. Do not force the reader to reconstruct that it is a radius from a formula – which is not located near the text we are talking about – or from the nomenclature used (the small r).

The reviewer suggests replacing the term “size” with “radius” throughout the text or explaining once and for all in line 19 that the authors consider the radius to be the characteristic size of the pore.

Line 22

“sensitive pore size”

As stated in the first review, it is good practice to explain the meaning of each term the first time this term is introduced. Here we are faced with a “Nomenclature” section, introduced without giving explanations on the meaning of the terms listed there. The reviewer understands the need to introduce the symbols that will be used in the text, but you also need to explain what these symbols represent.

The reviewer suggests introducing a few lines before the list of symbols or a footnote. The few lines (or the footnote) can be used to explain what a “sensitive pore” and a “sensitive pore size” are. Be aware that you provided a definition of “sensitive pore size” in line 117 (many lines after using “sensitive pore size” for the first time), but a “sensitive pore” is something other than a “sensitive pore size” and it must be defined separately.

Line 105

“which can well detect the porosity”

Consider changing this sentence: placed in this position, “which” refers to “concrete materials”, while it is quite evident that the verb “detect” should be related to the MIP method.

Line 160

“each pore size is an independent circle”

A size is a one-dimensional length, while a circle is two-dimensional. Therefore, a size cannot be a circle. The authors probably meant that “each pore is an independent circle”.

Author Response

Dear Editor,

Thank you very much for your letter and the comments from the referees about our manuscript, Study of concrete strength and pore structure model based on Grey Relation Entropy. We acknowledge your comments and constructive suggestions very much, which are valuable for improving the quality of our manuscript. We have provided itemized responses in blue colored text. And we have checked and revised the manuscript according to the reviewers’ comments in detail. All changes in the revised manuscript were highlighted using the yellow background for easy checking and editing. We hope, with these modifications and improvements based on your suggestions and the reviewers’ comments, the quality of our manuscript would meet the publication standard of Current Science. We submit here the revised manuscript as well as the response to reviewer comments of our manuscript. If you have any question about this paper, please don’t hesitate to let me know.

Once again, thank you very much for your help to our paper processing.

Sincerely  yours,

Junfeng Guan

School of Civil Engineering and Communication, North China University of Water Resources and Electric Power

Response to reviewer comments point by point

Reviewer #2:

The reviewer appreciates the effort made by the author to improve the text of the paper. However, some aspects still deserve further investigation.

1.Line 19

“average pore size”

The reviewer appreciates the explanation of the authors based on the formula of the average pore size and the correction made to the term “diameter”. However, the reviewer remains of the opinion that the term “size” is too general, as the characteristic size of an object assimilated to a sphere is not necessarily a radius. If it is a radius, it is good to call it “average pore radius”. Do not force the reader to reconstruct that it is a radius from a formula – which is not located near the text we are talking about – or from the nomenclature used (the small r).

The reviewer suggests replacing the term “size” with “radius” throughout the text or explaining once and for all in line 19 that the authors consider the radius to be the characteristic size of the pore.

 Many thanks for the nice suggestion.

Agrees with the suggestion very much, and have replaced the term “size” with “radius” throughout the text.

2.Line 22

“sensitive pore size”

As stated in the first review, it is good practice to explain the meaning of each term the first time this term is introduced. Here we are faced with a “Nomenclature” section, introduced without giving explanations on the meaning of the terms listed there. The reviewer understands the need to introduce the symbols that will be used in the text, but you also need to explain what these symbols represent.

The reviewer suggests introducing a few lines before the list of symbols or a footnote. The few lines (or the footnote) can be used to explain what a “sensitive pore” and a “sensitive pore size” are. Be aware that you provided a definition of “sensitive pore size” in line 117 (many lines after using “sensitive pore size” for the first time), but a “sensitive pore” is something other than a “sensitive pore size” and it must be defined separately.

 Many thanks for the nice suggestion.

 The question has been explained in the revised manuscript, the following content has been added.

“ .…….influence on the concrete's compressive strength than other factors in the micro-porous structure. The sensitive pore is the pore corresponding to the factor with the largest sensitivity coefficient. Because the radius of sensitive pores is within a range, the radius average value of the pores is the average pore radius of pore sensitive. In what follows it is assumed that the cross-sectional area of concrete sample is…….”

3.Line 105

“which can well detect the porosity”

Consider changing this sentence: placed in this position, “which” refers to “concrete materials”, while it is quite evident that the verb “detect” should be related to the MIP method.

Many thanks.

The question has been corrected in the revised manuscript.

“ .…….

The mercury intrusion porosimetry (MIP) method is used to test the pore characteristics of concrete materials, the MIP method can well detect the porosity, pore size distribution and other pore properties [31]. …….”

4.Line 160

“each pore size is an independent circle”

A size is a one-dimensional length, while a circle is two-dimensional. Therefore, a size cannot be a circle. The authors probably meant that “each pore is an independent circle”.

Many thanks.

The question has been corrected in the revised manuscript.

“each pore size is an independent circle” should be modified to “each pore is an independent circle”.

Reviewer 3 Report

Reviewers' comments:

Manuscript ID: materials-1051319

Title: Study of concrete strength and pore structure model based on Grey Relation Entropy.

Manuscript Type: Article.

The authors need to consider the following comments

- Please provides the references for all equations and formula.

- References: Make all references in same format for volume number, page number and journal name, because it is difficult to searching and reading (for example: [8] S. Zhang , K. Cao , C. Wang , et al. Influence of the porosity and pore size on the compressive and splitting strengths of cellular concrete with millimeter-size pores[J]. Construction and Building Materials, 2019, 235, [14] Y. Y. Huang. Concrete technology scientific theory foundation of China Concrete and Cement Products, 1985 (5), and [17] F. H.Wittmann. Structure of concrete with respect to crack growth. Fracture Mechanics of Concrete, 1983: 56-69).

- Several faults: are added or missing spaces between words: see manuscript file (for example: line numbers: 107, 171, 178, and etc..).

Author Response

Dear Editor,

Thank you very much for your letter and the comments from the referees about our manuscript, Study of concrete strength and pore structure model based on Grey Relation Entropy. We acknowledge your comments and constructive suggestions very much, which are valuable for improving the quality of our manuscript. We have provided itemized responses in blue colored text. And we have checked and revised the manuscript according to the reviewers’ comments in detail. All changes in the revised manuscript were highlighted using the yellow background for easy checking and editing. We hope, with these modifications and improvements based on your suggestions and the reviewers’ comments, the quality of our manuscript would meet the publication standard of Current Science. We submit here the revised manuscript as well as the response to reviewer comments of our manuscript. If you have any question about this paper, please don’t hesitate to let me know.

Once again, thank you very much for your help to our paper processing.

Sincerely  yours,

Junfeng Guan

School of Civil Engineering and Communication, North China University of Water Resources and Electric Power

Response to reviewer comments point by point

Reviewer #3:  

The authors need to consider the following comments

  1. Please provides the references for all equations and formula.

Many thanks.

The references have been provided in the revised manuscript for all equations and formula.

“ .…….

Then the sequence Y0(k) and Yi(k) is obtained. The correlation coefficient  is calculated according to Eq. (1) [29-30]:

                       (1)

.…….

The value of GRE for correlation coefficient series Ri(k) is calculated by Eq. (2) [29-30].

                       (2)

.…….

The relation degree of GRE D(xi) is calculated by Eq. (3) [29-30].

D(xi)= H(Ri) / Hmax                       (3)

where, Hmax is the maximum Grey Relation Entropy and Hmax=lnm, m is the number of data elements. 

.…….

as shown in Eq. (6) [18].

                         (6)

where, K2 is a test constant, which is mainly related to the strength of the concrete matrix, and is related to factors such as the type of aggregate, the type of cement and the cement content.

.…….

Balshin, Schiller, Ryshkewitch, Hansen and Jons [4-7] proposed several relationships between strength and porosity.

The linear relationship is given by Eq. (9) [4].

                       (9)

The power exponent relationship is given by Eq. (10) [5].

                       (10)

The exponential relationship is given by Eq. (11) [6].

                       (11)

The logarithmic relationship is given by Eq. (12) [7].

                       (12)

Atzeni et al. proposed the strength and pore structure relationship, as shown in Eq. (13) [16].

                       (13)

 . …….”

  1. References: Make all references in same format for volume number, page number and journal name, because it is difficult to searching and reading (for example: [8] S. Zhang , K. Cao , C. Wang , et al. Influence of the porosity and pore size on the compressive and splitting strengths of cellular concrete with millimeter-size pores[J]. Construction and Building Materials, 2019, 235, [14] Y. Y. Huang. Concrete technology scientific theory foundation of China Concrete and Cement Products, 1985 (5), and [17] F. H.Wittmann. Structure of concrete with respect to crack growth. Fracture Mechanics of Concrete, 1983: 56-69).

 Many thanks for the nice suggestion.

The question on the references has been corrected in the revised manuscript.

“ .…….

References

  • Zhang , K. Cao , C. Wang , et al. Influence of the porosity and pore size on the compressive and splitting strengths of cellular concrete with millimeter-size pores. Construction and Building Materials, 2020, 235:117508.
  • Liu , C. Liu , G. Bai, et al. Impact of chloride intrusion on the pore structure of recycled aggregate concrete based on the recycled aggregate porous interface. Construction and Building Materials, 2020, 259:120397.
  • Z. Xu, M. S. Tang. T. C. Some Problems in T. C. Powers' Model. Journal of the Chinese Ceramic Society, 1991 (3): 104-111. (in Chinese)
  • Chen, S. Wu, J Zhou. Influence of porosity on compressive and tensile strength of cement mortar. Construction & Building Materials, 2013, 40: 869-874.
  • M. Brandt. Cement-Based Composites: Materials, Mechanical Properties and Performance, Second Edition. Crc Press, 2009: 26-48.
  • C. Hansen. Physical Structure of Hardened Cement Paste. A Classical Approach. Materials and Structures, 1986, 19(6): 423-436.
  • Ryshkewitch. Compression Strength of Porous Sintered Alumina and Zirconia. Journal of the American Ceramic Society, 2010, 36(2): 65-68.
  • Guo , S. Wu , Z. Lyu , et al. Pore structure characteristics and performance of construction waste composite powder-modified concrete. Construction and Building Materials, 2020: 121262.
  • S. Jons, B. Osbaeck. The effect of cement composition on strength described by a strength-porosity model. Cement & Concrete Research, 1982, 12(2): 167-178.
  • S. Pann, T. Yen, C. W. Tang, et al. A New Strength Model Based on Water/Cement Ratio and Capillary Porosity. Aci Materials Journal, 2003, 100(4): 311-318.
  • Kolias. Investigation of the possibility of estimating concrete strength by porosity measurements. Materials and Structures, 1994, 27: 265-272.
  • Jambor. Pore structure and strength development of cement composites. Cement & Concrete Research, 1990, 20(6): 948-954.
  • W. Wu, H. Z. Lian. High Performance Concrete. Beijing: China Railway Publishing House, 1999. (in Chinese)
  • Y. Huang. Some progress in the scientific research of concrete materials (part 1). Concrete World, 2016(07): 35-39. (in Chinese)
  • Odler, M. Rößler. Investigations on the relationship between porosity, structure and strength of hydrated Portland cement pastes. II. Effect of pore structure and of degree of hydration. Cement & Concrete Research, 1985, 15(3): 401-410.
  • Atzeni, L. Massidda, U. Sanna. Effect of pore size distribution on strength of hardened cement pastes. Proceedings of the First International RILEM Congress on Pore Structure and Material Properties, Paris, 1987: 195-202.
  • H. Wittmann, K. Rokugo, E. Brühwiler, et al. Fracture energy and strain softening of concrete as determined by means of compact tension specimens. Materials and Structures, 1988, 21(1):21-32.
  • Kumar, B. Bhattacharjee. Porosity, pore size distribution and in situ strength of concrete. Cement & Concrete Research, 2003, 33(1): 155-164.
  • Tang. A study of the quantitative relationship between strength and pore-size distribution of porous materials. Cement & Concrete Research, 1986, 16(1): 87-96.
  • J. Zhang, X. Zhang. Grey correlation analysis between strength of slag cement and particle fractions of slag powder. Cement and Concrete Composites, 2007,29(6): 498-504.
  • Gao. Study on the relationship between air void structure and strength of concrete based on BP neural network. Construction Technology, 2017, S1(46): 292-295. (in Chinese)
  • Yao, L. Li, J. Guan, et al. Initial cracking strength and initial fracture toughness from three-point-bending and wedge splitting concrete specimens. Fatigue & Fracture of Engineering Materials & Structures, 2020, DOI: 10.1111/ffe.13381.
  • Shilpa, G. S. Prakash, M. R. Shivakumar. A combinatorial approach to optimize the properties of green sand used in casting mould. Materials today: proceedings, 2020.
  • Guo, T. Nian, F. Zhou. Analysis of factors that influence anti-rutting performance of asphalt pavement. Construction and Building Materials, 2020, 254: 119237.
  • M. Mokhtar, S. A. Abo-El-Enein, M. Y. Hassaan, et al. Mechanical performance, pore structure and micro-structural characteristics of graphene oxide nano platelets reinforced cement. Construction & Building Materials, 2017, 138: 333-339.
  • Jin, J. Zhang, S. Han. Fractal analysis of relation between strength and pore structure of hardened mortar. Construction & Building Materials, 2017, 135:1-7.
  • L. Chen, H. S. Yang, W. W. LI, et al. Micro-pore structures of MgO concrete at long-term age. Journal of Hydroelectric Engineering, 2016, 35(6): 118-124. (in Chinese)
  • Xie, Q. C. Wang, S. Li, et al. Relations of pore structure and compressive strength of concrete under different water to binder ratio and curing condition. Bulletin of Chinese Ceramic Society, 2015, 34(12):3695-3702. (in Chinese)
  • Wu, P. Li, T. Nian, et al. Evaluation of asphalt and asphalt mixtures' water stability method under multiple freeze-thaw cycles. Construction and Building Materials, 2019, 228(Dec.20):117089.1-117089.15.
  • Sanghvi, D. Vora, J. Patel et al. Optimization of end milling of Inconel 825 with coated tool: A mathematical comparison between GRA, TOPSIS and Fuzzy Logic methods. Materials Today: Proceedings, https://doi.org/10.1016/j.matpr.2020.06.413.
  • Berodier, J. Bizzozero, A. Muller. Mercury Intrusion Porosimetry. A Practical Guide to Microstructural Analysis of Cementitious Materials, 2015.

. …….”

3.Several faults: are added or missing spaces between words: see manuscript file (for example: line numbers: 107, 171, 178, and etc..).

Many thanks.

The question has been corrected in the revised manuscript.
